# Comparative functional survival and equivalent annual cost of 3 long-lasting insecticidal net (LLIN) products in Tanzania: A randomised trial with 3-year follow up

Lena M. Lorenz[1,2], John Bradley[3], Joshua Yukich[4], Dennis J. Massue[5,6,7,8], Zawadi Mageni Mboma[1,9], Olivier Pigeon[10], Jason Moore[6,7], Albert Kilian[11], Jo Lines[1], William Kisinza[5], Hans J. Overgaard[12,13‡], Sarah J. Moore[6,7,8‡*]

1 Department of Disease Control, Faculty of Infectious and Tropical Diseases, London School of Hygiene &Tropical Medicine, London, United Kingdom, 2 Queen's Medical Research Institute, College of Medicine and Veterinary Medicine, University of Edinburgh, Edinburgh, United Kingdom, 3 MRC Tropical Epidemiology Group, London School of Hygiene & Tropical Medicine, London, United Kingdom, 4 Department of Tropical Medicine, Tulane University School of Public Health and Tropical Medicine, New Orleans, Louisiana, United States of America, 5 National Institute for Medical Research, Amani Research Centre, Muheza, Tanzania, 6 Vector Control Product Testing Unit, Ifakara Health Institute, Bagamoyo, Tanzania, 7 Epidemiology and Public Health Department, Swiss Institute of Tropical and Public Health, Basel, Switzerland, 8 University of Basel, Basel, Switzerland, 9 Ifakara Health Institute, Dar es Salaam, Tanzania, 10 Plant Protection Products and Biocides Physico-chemistry and Residues Unit, Agriculture and Natural Environment Department, Walloon Agricultural Research Centre, Gembloux, Belgium, 11 Tropical Health, Montagut, Spain, 12 Faculty of Science and Technology, Norwegian University of Life Sciences, Ås, Norway, 13 Department of Microbiology, Faculty of Medicine, Khon Kaen University, Khon Kaen, Thailand

‡ These authors are joint senior authors on this work.
* smoore@ihi.or.tz

**Data Availability Statement:** The data underlying the results presented in the study are available

## Abstract

### Background

Two billion long-lasting insecticidal nets (LLINs) have been procured for malaria control. A functional LLIN is one that is present, is in good physical condition, and remains insecticidal, thereby providing protection against vector-borne diseases through preventing bites and killing disease vectors. The World Health Organization (WHO) prequalifies LLINs that remain adequately insecticidal 3 years after deployment. Therefore, institutional buyers often assume that prequalified LLINs are functionally identical with a 3-year lifespan. We measured the lifespans of 3 LLIN products, and calculated their cost per year of functional life, to demonstrate the economic and public health importance of procuring the most cost-effective LLIN product based on its lifespan.

### Methods and findings

A randomised double-blinded trial of 3 pyrethroid LLIN products (10,571 nets in total) was conducted at 3 follow-up points: 10 months (August–October 2014), 22 months (August–October 2015), and 36 months (October–December 2016) among 3,393 households in Tanzania using WHO-recommended methods. Primary outcome was LLIN functional survival (LLIN present and in serviceable condition). Secondary outcomes were (1) bioefficacy and

from Norwegian Centre for Research Data (NSD), Access to the data sets must be requested from NSD using this order form: https://nsd.no/nsd/english/order.html and referring to the project ID no. NSD2695 and project name "The useful life of bednets for malaria control in Tanzania: Attrition, Bioefficacy, Chemistry, Durability and insecticide Resistance", project no. 220757.

**Funding:** HJO Research Council of Norway 970141 669 https://www.forskningsradet.no/en. The funders had no role in study design, data collection and analysis, decision to publish, or preparation of the manuscript.

**Competing interests:** I have read the journal's policy and the authors of this manuscript have the following competing interests: The authors WK, JM, SJM and OP conduct product evaluations for the following companies that produce Long Lasting insecticidal Nets: A to Z, BASF, DCT, Life Ideas, LandScent, Moon Netting, Real Relief, Syngenta, Sumitomo, Vestergaard Frandsen, VKA Polymers, Yorkool. AK is employed by Tropical Health LLP that carries out evaluations of bednet durability.

**Abbreviations:** EAC, equivalent annual cost; HR, hazard ratio; IACT, Ifakara Ambient Chamber Test; LLIN, long-lasting insecticidal net; WHO, World Health Organization.

chemical content (residual insecticidal activity) and (2) protective efficacy for volunteers sleeping under the LLINs (bite reduction and mosquitoes killed). Median LLIN functional survival was significantly different between the 3 net products ($p = 0.001$): 2.0 years (95% CI 1.7–2.3) for Olyset, 2.5 years (95% CI 2.2–2.8) for PermaNet 2.0 (hazard ratio [HR] 0.73 [95% CI 0.64–0.85], $p = 0.001$), and 2.6 years (95% CI 2.3–2.8) for NetProtect (HR = 0.70 [95% CI 0.62–0.77], $p < 0.001$). Functional survival was affected by accumulation of holes, leading to users discarding nets. Protective efficacy also significantly differed between products as they aged. Equivalent annual cost varied between US$1.2 (95% CI $1.1–$1.4) and US$1.5 (95% CI $1.3–$1.7), assuming that each net was priced identically at US$3. The 2 longer-lived nets (PermaNet and NetProtect) were 20% cheaper than the shorter-lived product (Olyset). The trial was limited to only the most widely sold LLINs in Tanzania. Functional survival varies by country, so the single country setting is a limitation.

## Conclusions

These results suggest that LLIN functional survival is less than 3 years and differs substantially between products, and these differences strongly influence LLIN value for money. LLIN tendering processes should consider local expectations of cost per year of functional life and not unit price. As new LLIN products come on the market, especially those with new insecticides, it will be imperative to monitor their comparative durability to ensure that the most cost-effective products are procured for malaria control.

## Author summary

### Why was the study done?

- Over 2 billion long-lasting insecticidal nets (LLINs) have been procured for malaria control. Modelling has shown that longer-lasting LLINs would save stakeholders between US$500 million and US$700 million over a period of 5 years, yet LLIN tendering processes currently assume that all LLINs have the same lifespan.

- A functional LLIN must remain in the household, in good physical condition, and with adequate insecticidal activity to give good protection against malaria by preventing bites and killing mosquitoes.

- Before this study, only a few small studies in distinct geographical areas had compared the functional life of alternative LLIN products, mostly retrospectively.

- This 3-year randomised trial was designed to accurately compare the functional life of 3 leading LLIN brands, in order to help the Tanzanian government and other LLIN buyers to choose the most cost-effective LLINs.

## What did the researchers do and find?

- We randomised 3,393 households in Tanzania to 1 of 3 LLIN products (10,571 nets in total) and followed them for 3 years using methods recommended by the World Health Organization.

- This study showed that the functional life of LLINs in domestic use is less than 3 years and differs substantially between products. The main reason for different lifespans between brands was differential accumulation of physical damage that results in users discarding nets that they think are no longer protective. However, tests showed that all LLIN products were still partially protective against pyrethroid-susceptible mosquitoes after 3 years.

- In this trial, the most durable LLIN product was 20% more cost-effective (economic cost per year of effective life) than the least durable.

## What do these findings mean?

- Based on direct observation of a large number of nets in a range of study areas, our findings support previous studies suggesting that the functional life of LLINs may be less than 3 years.

- Our findings reveal that the lifespans of competing products can differ to a substantial and economically important degree.

- More durable LLINs would reduce the rate of loss of nets and the operational costs of malaria control, ultimately improving population access to this life-saving intervention.

- This study provides justification that measurement of the functional survival of new LLINs coming to market is an essential component of product evaluation for decision making. Functional survival affects LLIN cost; therefore, tendering processes should include a net durability component not just unit price.

## Introduction

The use of long-lasting insecticidal nets (LLINs) remains the most cost-effective way to control malaria and reduce mortality [1], notwithstanding insecticide resistance [2]. However, despite the procurement of 254 million LLINs in 2017 alone, global LLIN coverage remains inadequate, with only 56% of the population in endemic areas estimated to have access to a LLIN [3]. LLINs are mostly distributed through periodic mass distribution campaigns, and as a result, population access to LLINs fluctuates over time. Access is typically high directly after a mass campaign and then declines as nets wear out, often to 50% or less, until the next campaign. This fluctuating pattern of coverage, caused by nets wearing out, is seen across the African region [4], where gains in malaria control have stalled, and fewer than 50% of endemic countries remain on track to reach critical malaria reduction targets [3]. Investment in malaria control has stagnated and was US$2.3 billion (50%) below the resources required to meet the World Health Organization (WHO) targets of 40% reductions in malaria case incidence and

mortality rates by 2020 [5]. These gaps in funding and coverage emphasise the need to deploy products that present the best value for money.

A report to the Malaria Policy Advisory Committee (MPAC) advised that increasing the functional life of LLINs by 1 or 2 years would reduce the cost of malaria control by between US$500 million and US$700 million over a period of 5 years [6]. A functional LLIN is one that is present, is in good physical condition, and remains insecticidal, thereby providing protection against vector-borne diseases through preventing bites and killing disease vectors [6]. Durability, or functional survival, of LLINs varies between geographical regions [7] and environments [8,9] and remains an undervalued but critical determinant of the success and efficiency of malaria control programmes [10,11]. How long LLINs remain protective under user conditions will dictate how frequently they should be replaced, which has both public health and economic implications [12]. In 2011, it was calculated that in Tanzania, for mean LLIN lifespans of 2, 3, and 4 years, 89, 63, and 51 million LLINs, respectively, would be needed over 10 years to achieve national access targets [10].

Currently, WHO prequalifies products that demonstrate adequate insecticidal activity 3 years after deployment, but does not appraise the physical deterioration of nets over time as part of the LLIN prequalification assessment [13]. Historically, pyrethroid-treated LLINs were assessed in multi-country studies for physical and chemical durability over an anticipated lifespan of 3 years and 20 washes. In the mid-2000s, when these procedures were designed, we did not yet know the relative importance of attrition—the disappearance of nets from study households—as one of the main factors limiting the duration of protection from LLINs. Unfortunately, even after the importance of attrition had become very clear, the evaluation criteria were never changed to take account of it. Thus, of the nets tested in the current study, PermaNet 2.0 received WHO recommendation (now prequalification) based on pooled prospective data from 6 countries, where 80% of remaining nets met bioefficacy and net fabric integrity criteria [14]; Olyset received recommendation based on pooled retrospective data from 7 countries, where 77% of nets passed bioefficacy criteria, although net loss and damage could not be accurately assessed [15]; and NetProtect did not receive full recommendation due to inconsistencies in data between WHO-sponsored studies [16,17], and was withdrawn from the market after the trial reported here had started.

The WHO prequalification website lists a number of newly prequalified products as long-lasting (LLINs) [18], including some with active ingredients other than pyrethroids. The listing of these products was based on experimental hut data from 2 or 3 sites. Fabric integrity, residual chemical content, and bioefficacy data for products after operational household use through longitudinal studies or post-marketing surveillance are requested, but are not a requirement for prequalification. This has resulted in a tendering process where donors assume LLINs are identical, and procurement is weighted by the unit price of the commodity without regard to actual product lifespan [19]. However, all the available data suggest that the assumption of a uniform 3-year lifespan for all LLIN products is unrealistic [4]. There is a clear need for a more integrative economic approach, with purchasing decisions based on value for money and cost per effective unit of LLIN coverage [6,19]. New product classes of LLINs with novel active ingredients for insecticide resistance management are becoming available [20], but they remain susceptible to the same forces of physical disintegration, being discarded, and losing insecticidal activity. Moreover, in most cases, they are more expensive. This emphasises the need to consider the price of LLINs in terms of cost per year of functional life [12].

Here we report results from a large randomised trial of 3 LLIN products (PermaNet 2.0, Olyset, and NetProtect), conducted in 8 epidemiologically and ecologically distinct districts in Tanzania. The proportion of LLINs remaining in use and still protective against malaria mosquitoes was measured over 3 years of follow-up after deployment. We calculated relative LLIN

cost-effectiveness in terms of the equivalent annual cost (EAC), which is a conventional financial indicator used to compare products with different effective lifetimes. The median functional survival of each product and its EAC were calculated to inform optimal procurement of cost-effective LLINs.

## Methods

The trial has been described in detail previously [21]. It took place in 8 districts in Tanzania, selected to be representative of national environmental, ecological, and epidemiological settings (Fig 1). Within each district, 10 villages were randomly selected (except for Kinondoni [Dar es Salaam], where only 6 areas were available), and within each village, 45 households were recruited to participate in the trial. All households were randomised to receive 1 of 3 LLIN brands on a 1:1:1 ratio, stratified by village. The 3 brands were Olyset (manufactured with an enhanced knitting pattern that was introduced in 2013; permethrin incorporated in 150 denier polyethylene; Sumitomo Chemicals, Japan), PermaNet 2.0 (deltamethrin coated on 100 denier polyester; Vestergaard Frandsen, Switzerland), and NetProtect (deltamethrin incorporated in 110 denier polyethylene; BestNet, Denmark). Distribution of trial nets took place between October and December 2013. All nets owned by the participating households were collected and replaced with enough nets to cover all sleeping spaces. Before distribution, a sample of 10 nets per product was quality tested. Nets were the same size and colour and labelled by a 5-digit serial number so that participants and investigators remained blinded to the LLIN product until data collection was complete. In total, 3,393 households were randomised (1,132 to Olyset, 1,127 to PermaNet 2.0, and 1,134 to NetProtect), to which 10,571 nets were distributed.

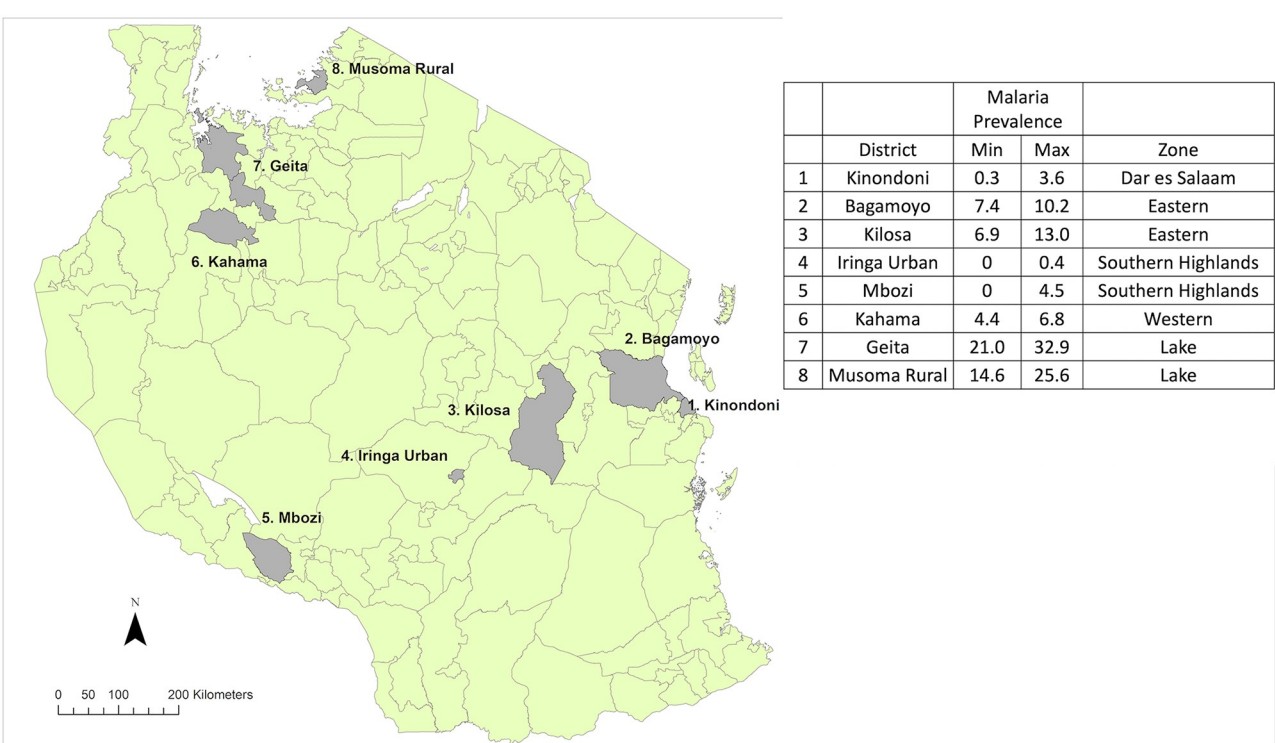

|  |  | Malaria Prevalence | |  |
|---|---|---|---|---|
|  | District | Min | Max | Zone |
| 1 | Kinondoni | 0.3 | 3.6 | Dar es Salaam |
| 2 | Bagamoyo | 7.4 | 10.2 | Eastern |
| 3 | Kilosa | 6.9 | 13.0 | Eastern |
| 4 | Iringa Urban | 0 | 0.4 | Southern Highlands |
| 5 | Mbozi | 0 | 4.5 | Southern Highlands |
| 6 | Kahama | 4.4 | 6.8 | Western |
| 7 | Geita | 21.0 | 32.9 | Lake |
| 8 | Musoma Rural | 14.6 | 25.6 | Lake |

**Fig 1. Map of trial districts with 2015 malaria prevalence data (percent of children aged 6–59 months diagnosed with malaria by rapid diagnostic test and microscopy)** [22]**.** Open-access shapefiles from https://www.nbs.go.tz/index.php/en/census-surveys/gis.

Surveys were conducted among all consenting trial households when the LLINs were distributed and at 3 follow-up points: 10 months (August–October 2014), 22 months (August–October 2015), and 36 months (October–December 2016) (S1 Table). The serial numbers of the nets, linked to household-identifying codes in a master list, enabled follow-up of each net at each time point. At each follow-up visit, information on each LLIN was collected, including whether the net was present in the house, whether the net was in use, and, if the net was not present, reasons why it was not present. Physical integrity of LLINs was measured on a random sample of 3 nets per household by counting the number, location and size of holes [13,23]. Socioeconomic variables and a household member roster were also recorded. Electronic data capture was used for all surveys.

In addition to the data collected as part of the household surveys, at each time point 48 LLINs from each brand were randomly sampled from the master list and returned to a laboratory in Bagamoyo, Tanzania, for bioefficacy and chemical analysis using standard WHO methods [13,23] and, additionally, the Ifakara Ambient Chamber Test (IACT) [24]. Households received new nets to replace those removed for destructive sampling. Once a house had been sampled, it was eliminated from the master list to prevent confounding of results. Table 1 describes the different components of LLIN durability, the tests conducted to obtain the data, the outcome indicators for statistical analysis, and the corresponding WHO threshold criteria [6,13,23]. The numbers of LLINs tested for each of the components of LLIN durability are listed in S1 Table.

**Table 1. LLIN durability components.**

| Component | Definition | Test conducted | Outcome indicators | WHO criteria or industry standard |
|---|---|---|---|---|
| Attrition | Net loss from household through discarding or alternative use | Household survey | Net presence | |
| Physical integrity | Physical state of the net to estimate bite protection | Count number, location, and size of hole(s) of a maximum 3 nets per household | Holed surface area measured by pHI [6] or MHSA ($cm^2$) | pHI 0–64, MHSA $\leq$ 79 $cm^2$: good |
| | | | | pHI 65–642, MHSA 80–789 $cm^2$: damaged |
| | | | | pHI $\leq$ 642, MHSA $\leq$ 789 $cm^2$: serviceable |
| | | | | pHI $\geq$ 643, MHSA $\geq$ 790 $cm^2$: too torn/unserviceable |
| Functional survival [6] | Estimation of nets still in households in serviceable condition | Median survival analysis | (Number of nets present and serviceable)/(number of nets originally received and not given away or lost to follow-up) | Median net survival in years = time point at which the estimate of functional survival crosses 50% |
| Biological efficacy | Ability of net to incapacitate or kill susceptible anopheline mosquitoes after contact with insecticide | IACT: whole nets [24] | Proportion of mosquitoes dead at 24 hours | |
| | | | Proportion of mosquitoes not blood fed | |
| | | WHO cone/tunnel test: 25 × 25 cm pieces [13] | Net samples meeting optimal bioefficacy criteria | 1-hour knock-down $\geq$ 95% or 24-hour mortality $\geq$ 80% or blood feeding inhibition $\geq$ 90% |
| Insecticide content | Amount of active ingredient in the net | Permethrin: GC-FID Deltamethrin: HPLC-DAD | Compliance of nets with WHO specifications at baseline; loss of active ingredient over time | Olyset: 20 g/kg ± 25% [15–25 g/kg] PermaNet: 1.4 g/kg ± 25% [1.05–1.75 g/kg] NetProtect: 1.8 g/kg ± 25% [1.35–2.25 g/kg] |

GC-FID, gas chromatography with flame ionisation detection; HPLC-DAD, high-performance liquid chromatography with diode array detection; IACT, Ifakara Ambient Chamber Test; LLIN, long-lasting insecticidal net; MHSA, median hole surface area; pHI, proportionate hole index; WHO, World Health Organization.

First, the protective efficacy of whole nets returned from the field was evaluated using IACT [24]. Each night, 10 male volunteers slept underneath 1 of the nets (or an untreated control net to monitor the quality of the bioassay) between 9 PM and 6 AM in a small chamber similar in size to a bedroom, within a screened compartment. At 9 PM, 30 laboratory-reared mosquitoes were released into the chamber. The next morning, all mosquitoes within the compartment were recaptured, and scored for 24-hour mortality and blood feeding inhibition. Each LLIN was tested twice on 2 consecutive nights. Subsequently, net pieces ($25 \times 25$ cm$^2$) were cut following the WHO sampling pattern and standard WHO cone bioassays were carried out [13]. If nets did not meet WHO optimal bioefficacy criteria for cone tests (Table 1), WHO tunnel tests were conducted [13]. All mosquito assays were conducted with fully pyrethroid-susceptible 2- to 8-day-old nulliparous female *Anopheles gambiae* sensu stricto (Ifakara strain). Insecticide content analyses were performed at Walloon Agricultural Research Centre (CRA-W) using standard Collaborative International Pesticides Analytical Council Limited (CIPAC) methods for determining LLIN insecticide content (Olyset, 331/LN/M/3; PermaNet 2.0, 333/LN/(M)/3; NetProtect, 333/LN/(M2)/3).

## Statistical analysis

All statistical analyses were conducted using Stata release 13 (StataCorp, College Station, TX). Data from the surveys at 10, 22, and 36 months were used to calculate attrition and functional survival (Table 1) using Kaplan–Meier estimators. For both attrition and functional survival, nets reported as given away, sold, or stolen were treated as lost to follow-up. Hazard ratios (HRs) for the difference in attrition and functional survival were calculated using discrete time survival analysis using a complementary log-log model [25]. Robust standard errors were used to account for the highest level of clustering (district) [26]. Of nets that were present, net condition was defined, following WHO recommendations, as 'good' or 'damaged' (combined as 'serviceable') or 'too torn/unserviceable' (Table 1). Negative binomial regression was used to compare hole surface area between net products. Data on WHO bioassays and the IACT test came from the 48 nets sampled at each time point. For WHO bioassays and the IACT test, if control mortality for an assay of a section of net was over 10%, the data from that section were not included in the analysis. A chi-squared test assessed the proportion of nets of each product passing the WHO bioefficacy criteria based on combined cone and tunnel tests. Logistic regression was used to analyse mortality and blood feeding inhibition from the IACT test; results were adjusted for chamber and experimental night, and robust standard errors were used to take account of nets being tested multiple times. A further analysis was conducted to test for differences in mortality between net brands in the IACT test based on net condition, in which net condition (defined above) was adjusted for as a fixed effect.

## Economic analysis

The EAC of an LLIN was calculated according to the standard formula [27]. To assess the value of longer functional survival, we used Eq 1, where *b* is the ratio of the lifespan of the more durable product to the lifespan of reference net *n*. The variable *r* is the discount rate. This relationship shows, for any change in net lifespan from *n* to *bn*, the relative increase in price, *a*, that would yield an identical EAC for the 2 products. Other factors being equal, a relative price increase less than *a* would favour the new, longer-lasting LLIN, while relative price increases greater than *a* would favour the reference net.

$$a = \frac{1 - (1+r)^{-bn}}{1 - (1+r)^{-n}} \tag{1}$$

Simulation of EACs for products tested in the trial was conducted using Monte Carlo methods, assuming a 3% discount rate, as is standard in health economic analysis. The baseline survival function for LLINs was estimated by regressing the survival proportions of Olyset nets derived from Kaplan–Meier analysis against time. The survival function was converted into a baseline hazard, and net failure lifetimes were simulated for a cohort of 500 LLINs assuming a Weibull distribution of time to failure (in terms of functional survival). The results of the cohort were summarised by estimating the median lifetime, and this process was repeated 10,000 times for each net type, yielding an estimate of the expected median lifetime and quantiles of its expected distribution. Results were converted into EACs with 95% quantiles. Distributional assumptions for the baseline hazard and the parameters of the Weibull distribution were fitted to the results. The baseline hazard and proportional hazard were simulated with log normal distributions (S2 Table).

## Ethics

Ethical approval was granted by ethical review committees at the London School of Hygiene & Tropical Medicine (6333/A443), Ifakara Health Institute (IHI/IRB/AMM/No: 07–2014), and the Tanzanian National Institute for Medical Research (NIMR/HQ/R.8c/Vol. I/285). Community sensitisation meetings were held prior to trial inception, and written informed consent was obtained from the head of the household or another adult household member of participating households before each survey. Volunteers for the IACT experiment were all Ifakara Health Institute staff members with appropriate training who gave written informed consent.

## Results

A total of 3,393 households were randomised, to which 10,571 nets were distributed (3,520 Olyset [33%], 3,513 PermaNet 2.0 [33%], and 3,538 NetProtect [33%]). The 3 trial arms were similar in number of participants, number of nets allocated, household characteristics, house design, and socioeconomic characteristics (Table 2). The proportion of households lost to follow-up was 20% over the 3 years of the trial.

### Functional survival

There were significant differences in functional survival (defined as presence of serviceable net) of the 3 products (Table 3). Estimated median functional survival was 2.0 years (95% CI 1.7–2.3) for Olyset, 2.5 years (95% CI 2.2–2.8) for PermaNet, and 2.6 years (95% CI 2.3–2.8) for NetProtect ($p < 0.001$). There was no significant difference in net use by net product (S3 Table).

### Economic analysis

Simulation results show that the expected EAC in US dollars of the 3 LLINs in the trial varied between $1.2 (95% CI $1.1–$1.4) for PermaNet and NetProtect and $1.5 (95% CI $1.3–$1.7) for Olyset, assuming that each net was priced identically at $3.0 (Table 3). The longer-lived net products (PermaNet and NetProtect) were approximately 20% lower in EAC than the shorter-lived Olyset product.

### Components of functional survival and secondary outcomes

**Attrition.** There were significant differences in attrition between net products. Olyset nets were lost at a faster rate than PermaNet 2.0 and NetProtect nets (Table 4). After 3 years, 55% of Olyset nets were no longer present in households, compared to 42% of PermaNet 2.0

**Table 2. Household and socioeconomic characteristics of participating households in each trial arm.**

| Characteristic | Olyset | PermaNet 2.0 | NetProtect |
|---|---|---|---|
| Number of nets distributed | 3,520 | 3,513 | 3,538 |
| Number of participants | 6,061 | 6,024 | 6,200 |
| Number of households | 1,132 | 1,127 | 1,134 |
| Average household size | 5.8 | 5.8 | 6.5 |
| Mean sleeping spaces per household | 3.65 | 3.55 | 3.55 |
| Mean nets per household | 2.92 | 2.96 | 3.04 |
| Male household members (%) | 49 | 48 | 49 |
| Female household members (%) | 51 | 52 | 51 |
| Age distribution of household members (%) | | | |
| ≤5 years | 16.64 | 17.21 | 17.56 |
| 6–17 years | 33.16 | 33.27 | 34.19 |
| 18–50 years | 37.61 | 39.16 | 37.73 |
| ≥51 years | 12.60 | 10.36 | 10.52 |
| Highest level of education of household head (%) | | | |
| No education | 21.62 | 19.99 | 20.69 |
| Some primary education | 30.23 | 29.26 | 20.69 |
| Completed primary school | 32.60 | 33.54 | 39.66 |
| Secondary education | 6.45 | 6.75 | 5.17 |
| Housing materials (%) | | | |
| Roof: thatch | 19.88 | 17.11 | 17.08 |
| Roof: tin | 79.89 | 82.60 | 82.56 |
| Walls: mud and sticks | 17.30 | 14.96 | 14.65 |
| Walls: mud brick | 24.15 | 21.81 | 22.18 |
| Walls: burned brick | 40.32 | 43.54 | 43.98 |
| Walls: cement brick | 18.23 | 19.69 | 19.19 |
| Floor: mud | 52.97 | 48.42 | 49.89 |
| Floor: cement | 43.17 | 46.13 | 44.48 |
| Socioeconomic quintile (%) | | | |
| 1 (least wealthy) | 21.90 | 18.99 | 19.23 |
| 2 | 20.59 | 19.06 | 20.60 |
| 3 | 19.85 | 20.12 | 20.29 |
| 4 | 19.70 | 20.65 | 19.52 |
| 5 (most wealthy) | 17.96 | 21.18 | 20.37 |

and 46% of NetProtect nets ($p < 0.001$; Table 4). Of the 10,571 nets distributed, 4,964 (46%) were lost over the whole trial period (S5 Table).

**Physical integrity.** The condition of nets that remained in households deteriorated over the course of the trial. At each time point, Olyset had the largest proportion and NetProtect had the smallest proportion of 'too torn' nets (Fig 2). The median hole surface area in Olyset nets increased from 38 cm$^2$ at 10 months to 459 cm$^2$ after 36 months, compared to 6 cm$^2$ to 295 cm$^2$ for PermaNet 2.0 and 8 cm$^2$ and 152 cm$^2$ for NetProtect (S6 Table). Questionnaire data showed that at 3 years, 70% of nets no longer in use had been discarded when they were perceived as too damaged to be useful. Others were given away (17%), stolen (3%), or repurposed (3%).

**Bioefficacy.** At baseline, all products met optimal WHO bioefficacy criteria. After field use, there were significant differences in the bioefficacy of the net products measured using standard WHO cone and tunnel tests over time (Table 5). At 10 months, 100% of NetProtect

**Table 3. Percentage net functional survival (defined as presence of the net in the house and in serviceable condition) and simulated equivalent annual cost (assuming S$3.0 purchase price) by net product and time point.**

| Net product | Percent functional survival (95% CI) | | | Median survival in years (95% CI)[†] | Hazard ratio (95% CI), p-value | Simulated equivalent annual cost in US dollars (95% CI) |
|---|---|---|---|---|---|---|
| | 10 months | 22 months | 36 months | | | |
| Olyset | 82 (79, 85) | 54 (47, 62) | 27 (20, 34) | 2.0 (1.7, 2.3) | 1 | 1.5 (1.3, 1.7) |
| PermaNet | 88 (85, 90) | 65 (57, 72) | 38 (31, 46) | 2.5 (2.2, 2.8) | 0.73 (0.64, 0.85), p = 0.001 | 1.2 (1.1, 1.4) |
| NetProtect | 88 (84, 91) | 67 (61, 72) | 40 (34, 45) | 2.6 (2.3, 2.8) | 0.70 (0.62, 0.77), p < 0.001 | 1.2 (1.1, 1.4) |
| | | | | | p = 0.001* | |

[†]Details of the survival analysis are provided in S4 Table.

*p-Value for the comparison between the 3 nets. For the difference between PermaNet and Netprotect, p = 0.199.

and PermaNet 2.0 nets met WHO optimal bioefficacy criteria, compared to 73% of Olyset nets (p < 0.001). Nets decreased in bioefficacy through time, but even after 3 years, 96% of NetProtect, 85% of PermaNet 2.0, and 75% of Olyset nets met WHO criteria for bioefficacy (p = 0.017; Table 5).

When whole nets were tested after 3 years using IACT, 88% of Olyset, 96% of PermaNet 2.0, and 92% of NetProtect nets passed WHO optimal criteria of ≥80% mortality and ≥90% blood feeding inhibition. There were differences between products in 24-hour mortality. Olyset showed lower mortality (p < 0.001), but all 3 products showed similar levels of feeding inhibition (Fig 3; S7 Table). Mosquito mortality was higher for nets defined as 'too torn' (odds ratio = 0.65 [95% CI 0.49–0.88], p = 0.005), and the differences in mosquito mortality between the net products remained significant after adjusting for physical condition. Similarly, protection from mosquito bites (feeding inhibition) was considerably lower among nets that were 'too torn' (OR = 0.12 [95% CI 0.08–0.18], p < 0.001), but the differences between the net products remained non-significant after adjusting for physical condition.

**Active ingredient content.** At baseline, 100% (10) of Olyset and PermaNet 2.0 and 50% (5) of NetProtect samples complied with their target doses of active ingredient (S8 Table).

At 10 months, 22 months, and 36 months, mean permethrin content in Olyset nets decreased to 16.2 g/kg, 14.8 g/kg, and 13.0 g/kg, corresponding to a loss of 20%, 27%, and 36% of the original dose, respectively. Mean deltamethrin content of PermaNet 2.0 nets decreased to 0.75 g/kg, 0.47 g/kg, and 0.40 g/kg, corresponding to a loss of 48%, 68%, and 72% of the original dose, respectively. Mean deltamethrin content of NetProtect nets decreased to

**Table 4. Percentage attrition (defined as net loss due to discarding or alternative use of nets) and hazard ratios after 36 months by net product and time point.**

| Net product | Percent attrition (95% CI) | | | Hazard ratio (95% CI), p-value |
|---|---|---|---|---|
| | 10 months | 22 months | 36 months | |
| Olyset | 7 (5, 8) | 25 (21, 29) | 55 (49, 61) | 1 |
| PermaNet | 5 (3, 6) | 20 (17, 24) | 42 (38, 46) | 0.71 (0.64, 0.79), p < 0.001 |
| NetProtect | 6 (4, 8) | 22 (18, 26) | 46 (43, 50) | 0.81 (0.71, 0.93), p = 0.008 |
| | | | | p < 0.001* |

Details of the analysis are provided in S5 Table. Number of nets remaining in households by time point: 10 months, 8,269 nets; 22 months, 6,324 nets; 36 months, 3,942 nets.

*p-Value for the comparison between the 3 nets. For the difference between PermaNet and NetProtect, p = 0.006.

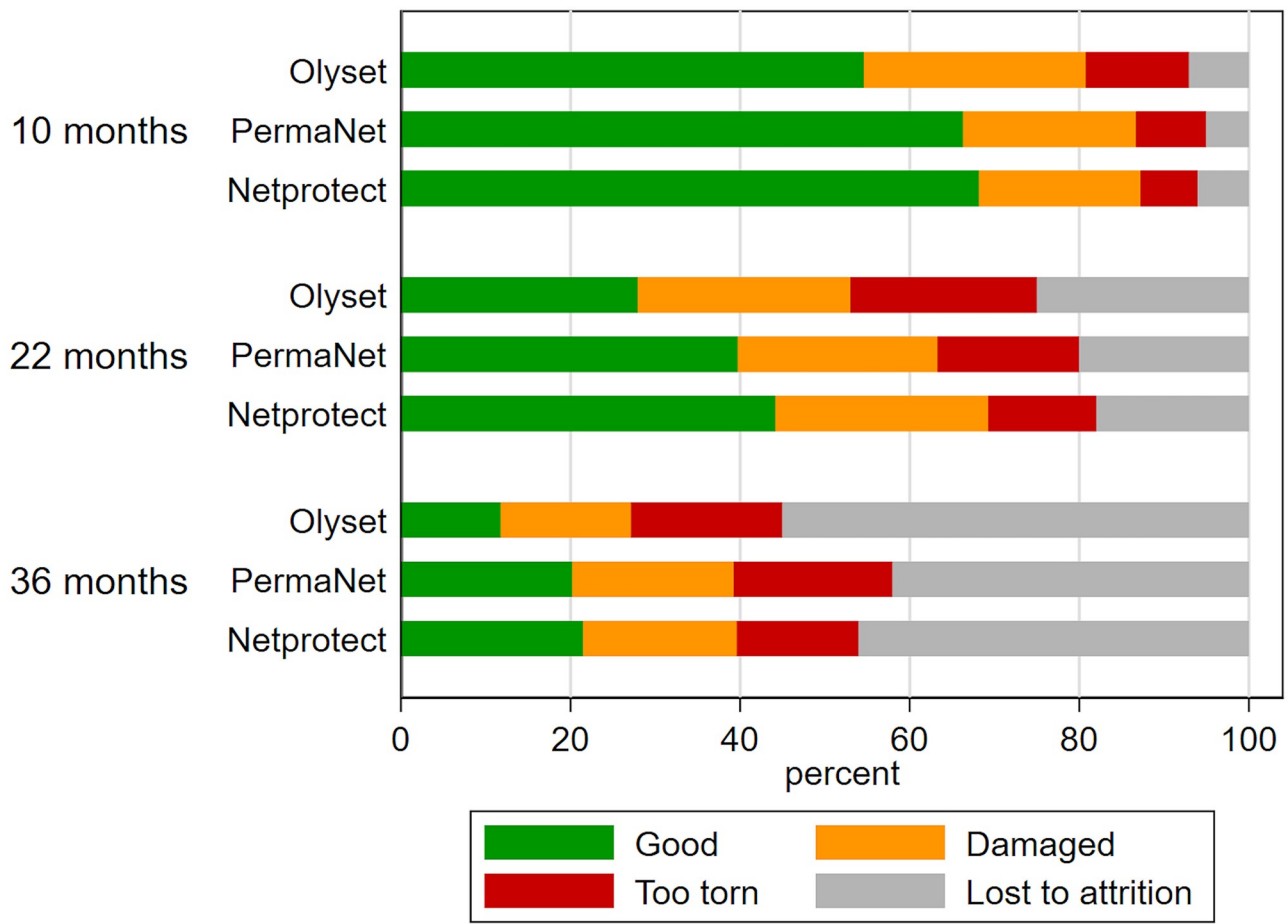

**Fig 2. Physical condition of long-lasting insecticidal nets remaining in households at time of survey according to WHO categorisation using proportionate hole index (pHI) [5] for the 3 net products and time points.** Green shows percent of nets in good condition (pHI 0–64), orange shows percent nets in damaged condition (pHI 65–642), and red shows percent of nets defined as 'too torn' (pHI ≥ 643). The sample sizes at 10 months were as follows: Olyset, 3,520; PermaNet, 3,513; NetProtect, 3,538. The sample sizes at 22 months were as follows: Olyset, 2,592; PermaNet, 2,622; NetProtect, 2,617. The sample sizes at 36 months were as follows: Olyset, 1,687; PermaNet, 1,827; NetProtect, 1,746.

0.91 g/kg, 0.52 g/kg, and 0.40 g/kg, corresponding to a loss of 33%, 61%, and 70% of the original dose, respectively (S8 Table). While this loss of insecticide did not negatively impact the bioefficacy of the nets against a pyrethroid-susceptible strain of mosquito, it is plausible that it would impact the efficacy of the nets against more resistant mosquitoes.

## Discussion

We conducted a randomised trial with 10,571 new LLINs of 3 brands (3,520 Olyset, 3,513 PermaNet, and 3,538 NetProtect) distributed among 3,393 households in 76 villages in 8 districts in Tanzania and followed up annually for 3 years. This was done to measure the rate at which the 3 net brands became damaged, lost bioefficacy, and were discarded by households. The findings of this trial demonstrate that there is considerable variability in the lifespan of pyrethroid-treated LLIN products. Our data also confirm that the median functional life of the LLINs in our study was less than 3 years in Tanzania, as also suggested by a systematic review of LLIN retention data in 39 sub-Saharan African countries [4]. A WHO-sponsored evaluation of NetProtect and PermaNet 2.0 conducted in Kenya showed very similar results to those

**Table 5. Percentages of net products meeting optimal WHO bioefficacy criteria by time point.**

| Net product | WHO cone test | | | WHO tunnel test | | | Overall (cone + tunnel) | | |
|---|---|---|---|---|---|---|---|---|---|
| | 10 months | 22 months | 36 months | 10 months | 22 months | 36 months | 10 months | 22 months | 36 months |
| Olyset | 4 | 8 | 14 | 72 | 78 | 71 | 73 | 79 | 75 |
| | (1, 14) | (2, 20) | (5, 27) | (57, 84) | (62, 89) | (54, 85) | (58, 85) | (65, 90) | (60, 87) |
| | [2/48] | [4/48] | [6/44] | [33/46] | [34/44] | [27/38] | [35/48] | [38/48] | [33/44] |
| PermaNet | 98 | 92 | 73 | 100 | 50 | 46 | 100 | 96 | 85 |
| | (89, 100) | (80, 98) | (58, 85) | (3, 100) | (7, 93) | (19, 75) | (92, 100) | (85, 99) | (72, 94) |
| | [46/47] | [44/48] | [35/48] | [1/1] | [2/4] | [6/13] | [47/47] | [46/48] | [41/48] |
| NetProtect | 100 | 100 | 73 | n/a | n/a | 85 | 100 | 100 | 96 |
| | (92, 100) | (93, 100) | (58, 85) | | | (55, 98) | (92, 100) | (93, 100) | (86, 99) |
| | [47/47] | [48/48] | [35/48] | | | [11/13] | [47/47] | [48/48] | [46/48] |
| | | | | | | | <0.001* | <0.001* | 0.017* |

95% confidence intervals in parentheses. Numbers passing/numbers tested in square brackets [*n/N*]. Nets are tested by cone test, and those that fail WHO optimal insecticide effectiveness criteria of ≥95% knock-down after 60 minutes or ≥80% 24-hour mortality are then further tested by tunnel test. Optimal criteria for the tunnel test are ≥80% 24-hour mortality or ≥90% blood feeding inhibition. Overall pass (cone and tunnel) is based on a net achieving 1 or more of these 4 criteria.

*$p$-Value for the comparison between the 3 nets. For the differences between Olyset and PermaNet, the $p$-values were <0.001, 0.014, and 0.208 at 10, 22, and 36 months, respectively. For the differences between Olyset and NetProtect, the $p$-values were <0.001, <0.001, and 0.004 at 10, 22, and 36 months, respectively. For the differences between PermaNet and NetProtect, the $p$-values were 1.0, 0.153, and 0.080 at 10, 22, and 36 months, respectively.

found here, with a median time to failure of 2.5 years for PermaNet 2.0 and 2.5 years for Net-Protect [16]. A full literature review of durability data available for the products evaluated in this trial is included in S1 Text. Summary net durability data available from peer-reviewed publications and WHO reports agree with the data in our trial for estimates of bioefficacy and fabric integrity after 3 years of operational use. The proportions of nets passing WHO bioefficacy criteria were above 80% for NetProtect and PermaNet 2.0 and slightly below 80% for Olyset. NetProtect and PermaNet had similar fabric integrity after 3 years of domestic use, with a higher proportion of serviceable nets relative to Olyset.

While there have been substantial economic investments to find new active ingredients, insecticide combinations, and synergists to combat the negative effects of insecticide resistance [28], the importance of durability for LLIN effectiveness has been side-lined. Consideration of its importance in vector control by key stakeholders such as the WHO may re-awaken the LLIN market to reward more durable products. This should, in turn, create incentives for investments in technological advances, research, and development by LLIN manufacturers [19]. There are indications that LLINs can be made substantially more durable for a small increase in unit price [29], and rapid technological evolution may be possible if there are appropriate market incentives.

The WHO's *Guidelines for Procuring Public Health Pesticides* [30] recommends that procurement decisions consider 'operational cost' rather than unit price, and an appropriate measure to compare value for money of LLINs would be 'cost per median year of net life under local conditions'. We measured the relative durability of nets using functional survival estimates, in terms of the EAC, and demonstrated that this approach outlined by WHO would indeed be useful. The cost analysis showed approximately 20% lower EAC when a longer-lasting LLIN (PermaNet 2.0 or NetProtect) was chosen over a shorter-lasting LLIN (Olyset), assuming prices for the products were identical. The economic modelling showed that the relative increase in price that is acceptable for a new product coming to market is also much smaller when the lifetime of the standard product increases (S1 Fig). Thus, the extension of the

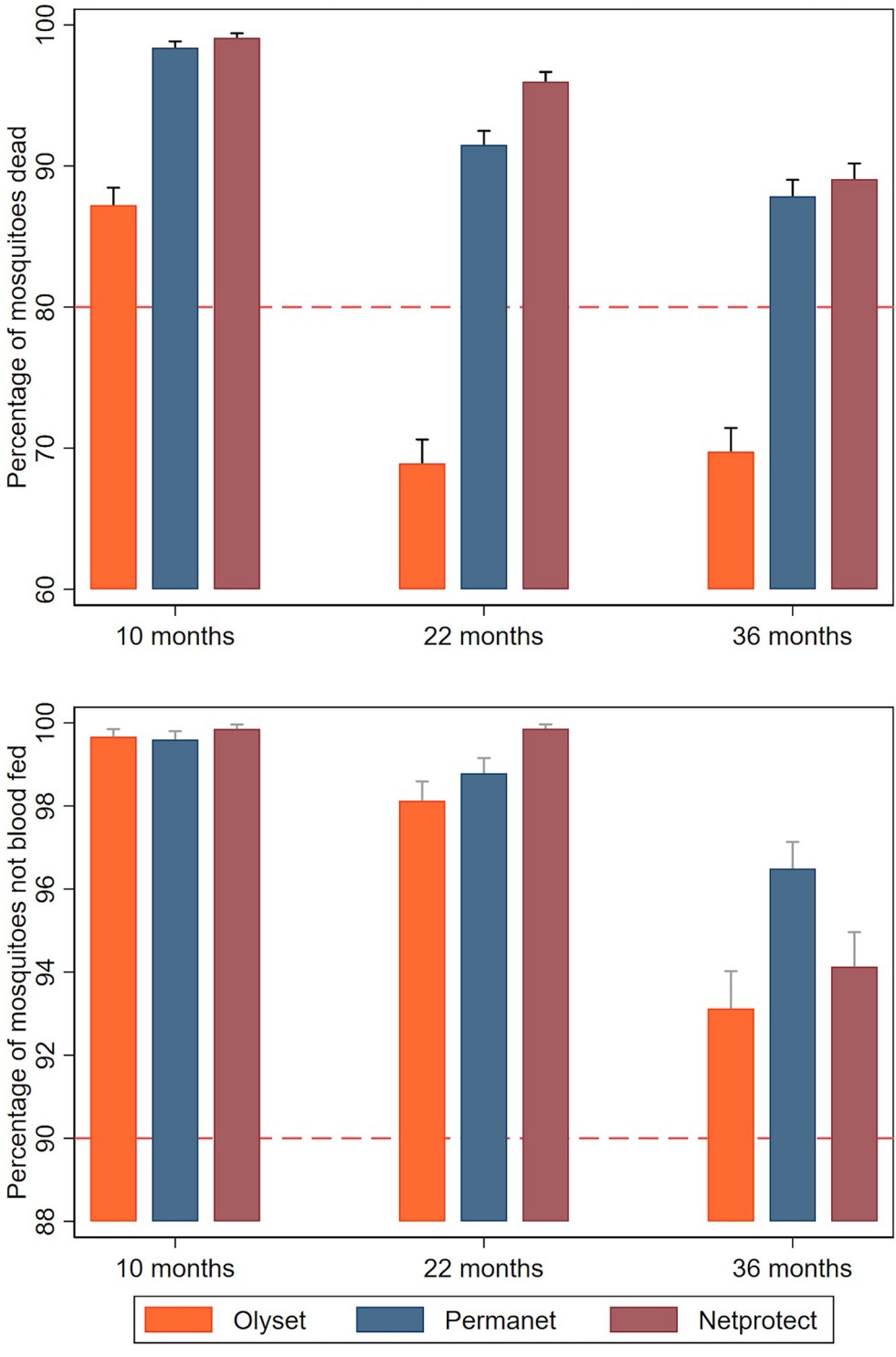

**Fig 3. Ifakara Ambient Chamber Test (IACT) results for mosquito mortality and blood feeding inhibition by net product and time point.** Mosquito mortality (top panel) and blood feeding inhibition (bottom panel). Orange, Olyset; blue, PermaNet; maroon, NetProtect. Optimal WHO criteria (80% mortality; 90% blood feeding inhibition) are indicated by dashed lines. The number of mosquitoes used at 10 months was as follows: Olyset, 2,700; PermaNet, 2,730; NetProtect, 2,730. The number of mosquitoes used at 22 months was as follows: Olyset, 2,880; PermaNet, 2,880; NetProtect, 2,880. The number of mosquitoes used at 36 months was as follows: Olyset, 2,880; PermaNet, 2,880; NetProtect, 2,880.

life of an innovator product is much more valuable if the standard product is relatively short-lived, as was seen in this study.

WHO requests LLIN manufacturers to provide data from 3 longitudinal field evaluations in different ecologies (e.g., West Africa, East Africa, and Asia) to retain prequalification listing. While it is recognised that durability is context-specific, we argue that it is possible to routinely generate median functional survival estimates and EACs for at least 3 locations using the WHO methodology outlined [13,23], albeit with a more limited sample size than the present study. The EAC may be a useful metric to compare cost-effectiveness of products, rather than the current practice of assessing products based simply on a minimum threshold of insecticidal activity after 3 years.

The limitation of the EAC metric is that it only captures the relative weighting of price and effective lifetime, while full cost-effectiveness and cost (including non-commodity costs) will result from a complex interaction of net durability, distribution modality, cost, and effectiveness. A limitation of the simplified approach here is that it does not fully consider these interactions, but it presents a straightforward and easily applicable approach to judging the relative cost and lifetime of a product.

Attrition and fabric integrity, the 2 factors that define physical survival of LLINs [6,31], differed significantly between the 3 net products. Olyset demonstrated more rapid accumulation of damage and faster attrition. In the current study and in previous work, we demonstrated that most LLINs were discarded because they were perceived by users as too damaged to offer protection against mosquito bites or malaria [32]. Attrition and fabric integrity are highly variable between contexts, and information on these factors is simpler to collect than bioefficacy or chemical content data. Further consideration should be given to developing simple tools to allow countries to assess attrition and fabric integrity during routine surveys (e.g., Malaria Indicator Surveys or Demographic and Health Surveys) to inform planning of intervals between mass distribution campaigns.

Of those nets still present after 3 years, 25%–40% were categorised as no longer physically serviceable, depending on the brand. However, even after 3 years, nets remained highly insecticidal when tested by bioassays against insecticide-susceptible malaria vectors. Damage actually increased the mortality of mosquitoes that entered nets through holes and became trapped, as also observed in other studies [33]. Indeed, torn LLINs continue to provide a degree of individual and community protection from malaria [34,35]. Our IACT experiments demonstrated that the 3 brands were all highly protective, although Olyset killed significantly fewer mosquitoes than PermaNet 2.0 and NetProtect. It is of note that the most common location for damage to the nets is on the bottom section of the nets at the point where they are tucked under a mat or mattress (S2 Fig). The act of tucking makes these holes inaccessible to mosquitoes even though the net appears to be badly damaged to the user, which may motivate them to discard the net.

A limitation of the trial is that only susceptible mosquitoes were used for bioefficacy testing. Pyrethroid resistance is widespread and increases feeding success and reduces mortality of mosquitoes [33]. Another limitation is the fact that the trial was only conducted in Tanzania (albeit in a wide range of epidemiological settings). Functional survival varies by country (S1 Text), so the single country setting is a limitation. However, the setting is more likely to affect absolute net survival rates than the comparison between LLIN products. Furthermore, the trial only included 3 brands of LLINs, all of which are treated with pyrethroids. As new LLIN products come on the market treated with different insecticides, insecticide combinations, or synergists, such as piperonyl butoxide (PBO), it will be imperative to monitor their comparative durability to ensure that the most cost-effective products are procured for malaria control. Functional life will have important implications for the selection of new products for resistance

management that have higher unit costs. New pyrethroid plus PBO nets may not be as durable as standard pyrethroid nets because PBO is lost rapidly from nets during washing, which reduces their efficacy [36]. However, in Tanzania, PBO nets continued to have superior public health benefits 2 years after distribution [20]. If the median functional survival of pyrethroid LLINs is 2 years, then PBO nets may remain cost-competitive.

Our findings confirm that even after 3 years, nets that are still in households, despite holes, still give partial protection against mosquito bites and continue to kill mosquitoes, providing some personal and community protection. However, if nets are discarded, or no longer used because they are perceived as too damaged, then they have no public health benefit at all. While it is possible to encourage users to retain their damaged, but still insecticidal, nets through behavioural change communication, a more effective and safer strategy would be to distribute more physically durable LLINs [29]. LLINs are the largest single cost item in the global malaria control budget. If LLIN effective lifespans became longer, net replacement needs would be substantially reduced, aiding in improving population access to this life-saving intervention despite the current stagnation in financial support for malaria control. It is technically feasible to manufacture more durable LLINs. However, this will happen only if institutional buyers consider cost-effectiveness for coverage [30] and give greater market share to longer-lasting and better value-for-money products.

## Supporting information

**S1 Fig. Relationship between increased net lifetimes in years and the acceptable increase in price.**
(TIF)

**S2 Fig. The location of damage on nets by year after distribution and net brand measured by proportionate hole index.**
(PDF)

**S1 STROBE Checklist.**
(PDF)

**S1 Table. Study flow.** The number of interviews completed each year, loss to follow-up, and the number of study nets evaluated for each durability component is shown.
(PDF)

**S2 Table. Parameters used in simulation of lifetimes for equivalent annual cost simulation analysis.**
(PDF)

**S3 Table. Reported net use the previous night by net product and time point.** Data represent numbers of respondents (percent) reporting use of nets.
(PDF)

**S4 Table. Number at risk (functional survival).**
(PDF)

**S5 Table. Number at risk (attrition).**
(PDF)

**S6 Table. Median hole surface area (in $cm^2$) and interquartile range (IQR) by net product and time point.**
(PDF)

**S7 Table. Ifakara Ambient Chamber Test (IACT) results for mosquito mortality and blood feeding inhibition by net product and time point (in months).**
(PDF)

**S8 Table. Number of nets, mean active ingredient (AI) content (g/kg), range (g/kg), and between net variation (%RSD); percentage of active ingredient lost over time; mean *R*-alpha isomer content (g/kg); and percentage of deltamethrin (only for PermaNet 2.0 and NetProtect) in net samples at baseline and 3 follow-up time points.**
(PDF)

**S1 Text. Literature review on durability of PermaNet 2.0, Olyset, and NetProtect nets.**
(PDF)

## Acknowledgments

Our special thanks are addressed to all technical staff at the Vector Control Product Testing Unit at Ifakara Health Institute for conducting data collection in the laboratory and semi-field experiments. We thank the LLIN manufacturers Sumitomo Chemical (Olyset), Vestergaard Frandsen (PermaNet), and BestNet (NetProtect) for their donation of the LLINs free of charge. Special thanks to Dr. Karen Kramer and Renate Mandike for their thoughtful comments that helped shape the trial design, and support at inception. This paper is published with the permission of the National Institute for Medical Research in Tanzania (NIMR/HQ/P12 Vol. XXVIII/20).

## Author Contributions

**Conceptualization:** Lena M. Lorenz, Jo Lines, Hans J. Overgaard, Sarah J. Moore.

**Data curation:** Lena M. Lorenz, John Bradley, Joshua Yukich, Hans J. Overgaard.

**Formal analysis:** John Bradley, Joshua Yukich, Olivier Pigeon.

**Funding acquisition:** Lena M. Lorenz, Hans J. Overgaard, Sarah J. Moore.

**Investigation:** Lena M. Lorenz, Dennis J. Massue, Zawadi Mageni Mboma, Olivier Pigeon, Jason Moore, Sarah J. Moore.

**Methodology:** Lena M. Lorenz, John Bradley, Joshua Yukich, Dennis J. Massue, Zawadi Mageni Mboma, Olivier Pigeon, Jason Moore, Albert Kilian, Jo Lines, William Kisinza, Hans J. Overgaard, Sarah J. Moore.

**Project administration:** Lena M. Lorenz, Dennis J. Massue, Zawadi Mageni Mboma, Jason Moore, William Kisinza, Hans J. Overgaard, Sarah J. Moore.

**Resources:** Hans J. Overgaard, Sarah J. Moore.

**Supervision:** Lena M. Lorenz, Jason Moore, Albert Kilian, Jo Lines, William Kisinza, Hans J. Overgaard, Sarah J. Moore.

**Visualization:** John Bradley, Joshua Yukich.

**Writing – original draft:** Lena M. Lorenz, John Bradley, Joshua Yukich, Olivier Pigeon, Jo Lines, Sarah J. Moore.

**Writing – review & editing:** Lena M. Lorenz, John Bradley, Joshua Yukich, Dennis J. Massue, Zawadi Mageni Mboma, Olivier Pigeon, Jason Moore, Albert Kilian, Jo Lines, William Kisinza, Hans J. Overgaard, Sarah J. Moore.

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
