## [Decision Letter · Decision Letter 0]

30 Mar 2020

Dear Dr. Moore,

Thank you very much for submitting your manuscript "Comparative functional survival and equivalent annual cost of three long lasting insecticidal net (LLIN) products in Tanzania: a three-year prospective cohort study of LLIN attrition, physical integrity, and insecticidal activity" (PMEDICINE-D-19-02821) for consideration at PLOS Medicine. 

Your paper was evaluated by a senior editor and discussed among all the editors here. It was also discussed with an academic editor with relevant expertise, and sent to three independent reviewers, including a statistical reviewer. The reviews are appended at the bottom of this email and any accompanying reviewer attachments can be seen via the link below:

[LINK]

In light of these reviews, I am afraid that we will not be able to accept the manuscript for publication in the journal in its current form, but we would like to consider a revised version that addresses the reviewers' and editors' comments. Obviously we cannot make any decision about publication until we have seen the revised manuscript and your response, and we plan to seek re-review by one or more of the reviewers. 

We expect to receive your revised manuscript by Apr 20 2020 11:59PM. Please email us (plosmedicine@plos.org) if you have any questions or concerns.

We look forward to receiving your revised manuscript. 

Sincerely,

Thomas McBride, PhD

Senior Editor 

PLOS Medicine

plosmedicine.org

1- I think the title can be shortened: “Comparative functional survival and equivalent annual cost of three long lasting insecticidal net products in Tanzania: a three-year prospective cohort study”

2- Thank you for listing the authors’ competing interests. Please list, either in the CI statement or in a supplemental file if the list is very long, the manufacturers that the authors work with.

3- Thank you for agreeing to make the data available and providing the link for data access. Please also provide any accession number(s) or search terms a researcher will need to obtain access to this specific dataset.

4- The Abstract Background is a bit long, please shorten by about half. 

5- Please amend “We conducted the largest prospective study of LLIN…” with “to our knowledge” or similar. 

6- Please include full date ranges for the study in the Abstract Methods and Findings section.

7- In the Abstract Methods and Findings section, please include both 95% CIs and p values for the study outcomes, including the cost estimates.

8- Please delete the final sentence of the Abstract Methods and Findings section (“However, the setting is more likely to...”).

9- Please preface the Abstract Conclusions with: “These results suggest…” or similar.

10- At this stage, we ask that you include a short, non-technical Author Summary of your research to make findings accessible to a wide audience that includes both scientists and non-scientists. The Author Summary should immediately follow the Abstract in your revised manuscript. This text is subject to editorial change and should be distinct from the scientific abstract. Please see our author guidelines for more information: https://journals.plos.org/plosmedicine/s/revising-your-manuscript#loc-author-summary

11- Thank you for providing the STROBE checklist. Please replace the line numbers with paragraph numbers per section (e.g. "Methods, paragraph 1"), since the line numbers of the final published paper may be different from the page numbers in the current manuscript.

12- Please remove the registered trademark symbols throughout the manuscript.

13- S1 Table could be moved to the main text.

14- Please begin the Discussion with a brief summary of what was done, before getting to the findings and interpretations.

15- The current first sentence of the Discussion would be better stated: “The findings of this study demonstrate that there is considerable variability in the lifespan of pyrethroid-treated LLIN products.”

16- On line 317, perhaps “Our findings demonstrate that nets that are still in households…”

Comments from the reviewers:

Reviewer #1: Although at first glance, this study may not appear particularly novel, and the results do not yield any major surprises it is an important paper that raises some fundamental points that urgently need to be addressed by all involved with bednet manufacturer, procurement and distribution.

As the authors point out, long lasting insecticidal nets (LLINs) represent a huge market with billions having been distributed across Africa in the 21st century. To be classified as 'long lasting' these nets must last three years under operational use and the World Health Organisation have issued guidelines on how these nets should be evaluated. But such prospective studies are rarely conducted in practice and surprisingly little has been published on the effective life of LLINS, particularly considering the billions of dollars spent on these. This paper looks at this specific issue in a study across multiple districts in Tanzania. In this particular study they find that one brand of net in inferior compared to two further nets tested. This is an important result for Tanzania and arguably for other countries too; although some might argue that the results will be context specific, as the authors point out it seems reasonable to expect the relative performance to remain.

The paper is weakened by not testing any of the net samples on resistant mosquitoes; this point is acknowledged by the authors but it cannot be overlooked. The vast majority of Anopheles mosquitoes in Tanzania, and across Africa, will have some level of resistance to pyrethroid insecticides. When nets are initially distributed they may still have sufficient insecticide to kill both susceptible and resistant mosquitoes but as the amount of active ingredient (AI) decreases (as observed in this study where two of the products have lost a fifth or a third of the AI within 10 months of use) it is possible that resistant mosquitoes will survive a short exposure to these nets and hence the operational efficacy of the nets will be compromised. The failure to measure the bioefficacy against resistant mosquitoes is a major limitation, particularly given it may have altered the key conclusions of the paper. 

I believe the paper could be strengthened in the following ways

1. It is essential to provide greater clarity how these nets were originally classified as LLINs. In the introduction (lines 27-29) the process is briefly described but, to evaluate the results of this manuscript, it is important that readers are aware of the specific evaluations that these three net products have undergone previously. This is important to understand if the current results are an anomaly or agree with previous studies. In the discussion this is mentioned again (line 275-277) but again no information on the previous evidence warranting the WHO classification of LLIN is provided. I would like to see a supplementary file, providing a summary of the durability data used by WHO to evaluate the three net types included in the current study, plus any subsequent published data on the durability of these nets (the authors don't even mention whether these results agree with their own retroactive study in Tanzania (Ref 26)). Furthermore, a fuller discussion of the extent to which WHO PQ are following these guidelines in their listings of new nets (particularly the new LLINs which contain more than one insecticide) must be included in the discussion. I agree with the authors conclusions that durability is too often overlooked but the extent to which this is happening, and the potential consequences of this could be more clearly stated.

2. If I remember correctly the weave design of Olyset changed at some point in the last six years; given the poor fabric integrity for Olyset reported it is critical to clarify if nets used here are those currently on the market.

3. Methods: The sampling strategy requires clarification

a. Were the three nets sampled per household pre-selected before entering (as opposed to convenience sampling on arrival)

b. When nets were sampled for bioeffiacy assays, was the net replaced? If so was the replacement clearly labelled to avoid being sampled in subsequent rounds?

4. Figure 2 legend should make clear what the denominator is (nets remaining in household at time of survey?) and indicate sample size for each row. This figure gives a somewhat misleading picture as it ignores nets that have been removed from households, many of which might be too torn. An additional (or replacement) figure including nets lost to follow up would be informative.

5. Figure 3 requires sample sizes. Please also change y scale from 0-100 as the current graph presents a misleading picture on the difference between the nets.

6. The results section 'Active ingredient content' is confusing for non-chemists. What is the biological significance of the R-alpha content? I assume this is not bioactive? The very high rate of loss of AI in two of the nets is not discussed further. This does not appear to relate to bioefficacy against susceptible mosquitoes but as the study did not test resistant strains, it is not possible to state whether this loss of insecticide would affect their operational efficacy.

7. Discussion, line 282, states that attrition and physical integrity are the two factors that define functional survival of LLINs - again I would query whether this is the case if you considered pyrethroid resistance. Churcher et al, e-Life 2018 have shown how net durability is influenced by resistance. This need acknowledging 

8. There was a national distribution of LLINs in Tanzania from 2015-2017, was the current study area included and, if so, how did this impact on the current study?

9. Discussion, line 253, states that median functional lifespan was closer to 2 years. However Table 2 states the median life span to be 2.0, 2.5 and 2.6 for the three nets - is this really closer to 2 years? A more accurate statement could be obtained if the lifespan of the pooled analysis was presented. If it is closer to 2.5 years rather than 2 years, state this. Six months is important in this context!

10. The analysis of the IACT tests mentions adjustment for physical condition. Please include further explanation of how this was performed in the data analysis section.

11. Discussion, line 299 'most damage to the nets is accumulated on the bottom section of the nets….'. Is this from this study? I couldn't find any information on the location of holes from the supplementary data tables. If not from this study a reference must be given. 

Reviewer #2: Statistical review

This paper reports a randomised study comparing the longevity and laboratory-assessed efficacy of mosquito nets. The authors show there are significant differences between the different types in both of these characteristics as assessed by different outcomes.

I had some comments on the statistical methods and reporting, which I have provided below.

1. Title: I was not sure why the title refers to this as a prospective cohort study, but the rest of the paper refers to it as a RCT - I would recommend this is kept consistent.

2. Abstract and results - I think it would be useful if confidence interval in differences of net functional survival was given between groups together with CIs and p-values; currently unclear (at least in the abstract) if the differences here are significant or not.

3. I did not find it completely clear from the methods section which outcomes were assessed by which data source (i.e. the inspection of the house or the lab tests on the restricted number). From what I understand, attrition, physical integrity and functional survival come from the household inspections and biological efficacy/insecticide content from the lab tests - please clarify if this is wrong.

4. Line 110 "adjusted for control mortality" - I didn't follow what exactly is being adjusted for here.

5. Methods - how was missing data handled in the analysis?

6. Statistical/economic analysis methods - was a pre-specified analysis plan developed for this analysis? It should be provided if so.

7. Page 10 line 159: is the p-value here testing if there's any difference between the three types of net? If not, could it be clarified what the p-value is testing?

8. Table 3 - is there any significant difference between permanet and netprotect here?

9. Table 4 - similarly here, it doesn't appear that permanet or netprotect would be significantly different?

James Wason

Reviewer #3: The manuscript presented by Lorenz et al. on the comparative functional survival and equivalent annual cost of three long-lasting insecticidal nets (LLIN) is clearly the best systematic assessment of LLIN durability to-date. The issue of how long LLINs last in the field is rapidly becoming the top priority topic for LLIN programmess, allongside insecticide resistance. Given that public entities spend in excess of USD 250 millions per year on LLIN deployment, and given that this represents our primary malaria prevention measure in most malaria-endemic settings, the importance of any major factor affecting the performance of LLINs can not be over-emphasized. 

The authors present here a well designed and well powered study comparing directly and in a randomized design the three brands of LLINs which largely dominate the market. These results are eagerly awaited by the malaria community and it is important to publish them rapidly. In addition to these important results, this manuscript also presents a number of new approaches to quantifying the issue of uselife of nets (or validates their use on a large scale), as well as a methodologically appropriate way to estimate the cost versus the expected effects of a product on the basis of its field characteristics. As a result of the above, I consider this a manuscript of high public health importance. 

The methodology of this work is largely sound and well implemented, and the results are well analyzed and described. The article is also well written and nearly all required information is there. I have therefore only one issue for which clarification is required.

1. Economic analysis (p 9 and 11). The authors have done the economic analysis on the basis of a uniform net price of USD 3. They should better justify their decision to take this price value for two reasons: (1) most factory prices of nets are lower than that amount these days, and (2) This price does not include the distribution costs as well as all the non-commodity costs associated with net distribution, such as Behaviour Change components, etc. The distribution costs (planning, census, transport, etc.) are particularly important to consider in this work because a net lasting longer would allow to lengthen the time interval between distributions, and that part of the total LLIN-in-the-field cost would be reduced proportionally. I appreciate that the authors have probably not taken this into account in their calculations, but they should at least discuss this point and outline what the wider implications are of an expended durability of LLINs in the field. 

Finally, the limitations of the study are well described, once the points above are included.

[LINK]

---

## [Decision Letter · Decision Letter 1]

18 Jun 2020

Dear Dr. Moore,

Thank you very much for re-submitting your manuscript "Comparative functional survival and equivalent annual cost of three Long Lasting Insecticidal Net (LLIN) products in Tanzania: a three-year prospective cohort study" (PMEDICINE-D-19-02821R1) for review by PLOS Medicine.

I have discussed the paper with my colleagues and the academic editor and it was also seen again by xxx reviewers. I am pleased to say that provided the remaining editorial and production issues are dealt with we are planning to accept the paper for publication in the journal.

[LINK]

We look forward to receiving the revised manuscript by Jun 25 2020 11:59PM. 

Sincerely,

Thomas McBride, PhD

Senior Editor 

PLOS Medicine

plosmedicine.org

Requests from Editors:

Comments from Reviewers:

Reviewer #1: The authors have made the requested additions/clarifications to the manuscript. 

Reviewer #2: Thank you to the authors for addressing my previous points. 

My only remaining minor point is related to my previous point 1: there are still references that make the study appear to be a randomised trial: the first line of the methods and findings of the abstract, together with other references to randomised allocation of the nets, together with descriptions of the study as a 'trial' in several places. I'm not sure why it is described to be an observational study when assignment is randomised. Perhaps a line somewhere in the methods would be useful to explain.

[LINK]

---

## [Editor Report · Decision Letter 2]

17 Aug 2020

Dear Dr. Moore, 

On behalf of my colleagues and the academic editor, Dr. Elizabeth Ashley, I am delighted to inform you that your manuscript entitled "Comparative functional survival and equivalent annual cost of three long lasting insecticidal net (LLIN) products in Tanzania:  a randomised trial with three-year follow up" (PMEDICINE-D-19-02821R2) has been accepted for publication in PLOS Medicine. 

PRODUCTION PROCESS

PRESS

PROFILE INFORMATION

Thank you again for submitting the manuscript to PLOS Medicine. We look forward to publishing it. 

Best wishes, 

Thomas McBride, PhD

Senior Editor 

PLOS Medicine

plosmedicine.org